TECHNICAL RELEASE

# PhysiCell Studio: a graphical tool to make agent-based modeling more accessible

Randy Heiland[1], Daniel Bergman[2,3], Blair Lyons[4], Grant Waldow[5], Julie Cass[4], Heber Lima da Rocha[1], Marco Ruscone[6,7,8,9], Vincent Noël[6,7,8] and Paul Macklin[1,*]

1 Department of Intelligent Systems Engineering, Indiana University, Bloomington, IN, USA
2 Department of Oncology, Sidney Kimmel Comprehensive Cancer Center, Johns Hopkins University, Baltimore, MD, USA
3 Convergence Institute, Johns Hopkins University, Baltimore, MD, USA
4 Allen Institute for Cell Science, Seattle, WA, USA
5 University of Wisconsin, Madison, WI, USA
6 Institut Curie, Université PSL, F-75005, Paris, France
7 INSERM, U900, F-75005, Paris, France
8 Mines ParisTech, Université PSL, F-75005, Paris, France
9 Sorbonne Université, Collège Doctoral, F-75005, Paris, France

**Submitted:** 27 January 2024

\* Corresponding author. E-mail: macklinp@iu.edu

Preprint submitted at https: //doi.org/10.1101/2023.10.24.563727

Included in the series: *PhysiCell Ecosystem* (https://doi.org/10.46471/ GIGABYTE_SERIES_0003)

## ABSTRACT

Defining a multicellular model can be challenging. There may be hundreds of parameters that specify the attributes and behaviors of objects. In the best case, the model will be defined using some format specification – a markup language – that will provide easy model sharing (and a minimal step toward reproducibility). PhysiCell is an open-source, physics-based multicellular simulation framework with an active and growing user community. It uses XML to define a model and, traditionally, users needed to manually edit the XML to modify the model. PhysiCell Studio is a tool to make this task easier. It provides a GUI that allows editing the XML model definition, including the creation and deletion of fundamental objects: cell types and substrates in the microenvironment. It also lets users build their model by defining initial conditions and biological rules, run simulations, and view results interactively. PhysiCell Studio has evolved over multiple workshops and academic courses in recent years, which has led to many improvements. There is both a desktop and cloud version. Its design and development has benefited from an active undergraduate and graduate research program. Like PhysiCell, the Studio is open-source software and contributions from the community are encouraged.

**Subjects** Software and Workflows, Cell Biology, Systems Biology

## INTRODUCTION AND BACKGROUND

Agent-based simulation frameworks [1] offer various approaches to modeling biological systems. PhysiCell [2] models cells as agents with independent attributes (e.g., position, volume, cycle status) and phenotypic behaviors (e.g., adhesion/repulsion, motility, secretion). PhysiCell is written in C++ and a model's parameters are defined using the eXtensible Markup Language (XML). As PhysiCell has evolved, many model parameters that were originally defined in C++ have been moved into XML. While this has been a definite improvement for modifying parameters during a model's development, it still poses significant challenges. Any moderately complex model now requires a rather large XML file, which makes it challenging to edit by hand. Some would argue that XML should not even be

edited by humans – that it was created primarily to be just a "machine-readable" (and editable) format. Unless a user is familiar with a text editor that supports XML syntax and can, for example, collapse sections of hierarchical information, it is difficult to see the skeleton of a PhysiCell model (its substrates and cell types) and visually associate parameters with their parent objects.

We present PhysiCell Studio, a graphical tool that makes it easier to build, run, and visualize a PhysiCell model. PhysiCell Studio began as a graphical user interface (GUI) that focused solely on editing the contents of the XML model. Over time, it has evolved to include additional functionality. A GUI can provide several benefits over a command line interface. This is especially true for a simulation framework like PhysiCell, where output results are visual as the scientist user interactively develops a model – changing parameters, running a simulation, plotting results, and repeating.

Benefits of using a GUI include: (1) easier to use, since point and click to access and edit objects and parameters offers an alternative to traditional text editing and is especially appealing to those who are less experienced developing code; (2) faster prototyping, where if the tool can also run a simulation and plot results, it can "close the loop", allowing for faster model development (and these capabilities can also allow users to skip setting up a development environment, which can be a barrier to getting started); and (3) reduce input errors, since a GUI can incorporate validation constraints, such as numeric input or pre-defined object selection.

PhysiCell Studio now joins other agent-based modeling frameworks that provide some level of GUIs, such as NetLogo [3], Chaste [4], Morpheus [5], CompuCell3D [6], Artistoo [7], and more.

## PHYSICELL MODELS

Defining a model in PhysiCell has been an evolving process as new functionality has been added over the past few years. A PhysiCell model currently consists of: (1) an XML configuration file containing model parameter values; (2) optional files (specified in the configuration file) that contain additional input data: initial conditions for cells (and in the future, substrates), and rules defining how cells respond to signals; and (3) optional custom C++ code. An executable model is the result of compiling the core PhysiCell C++ code together with any custom C++ code. Several sample models are provided in the PhysiCell source code distribution.

We show a portion of an XML configuration file in Figure 1. This is taken from one of PhysiCell's sample models ("interaction" model). We show just a single "cell_definition" (cell type) and its phenotype (containing more than 100 actual parameters). Note that for each of the eight phenotypic behaviors, we have collapsed the actual parameters and their values. There are seven cell types – "cell_definition" sections – in this particular model, bringing the total number of parameters for all cell types to more than 700. There will typically be multiple substrates (signals) defined in a model as well. For example, a COVID-19 PhysiCell model has 8 cell types and 11 substrates [8] (see: nanohub.org/tools/pc4covid19).

## BUILDING A TUMOR MODEL

We demonstrate PhysiCell Studio by showing how one could interactively build, explore, and iteratively refine a 2D tumor model. As is recommended when starting a PhysiCell model, we will load an existing "template" model from the sample projects, that is included

```
149    <cell_definitions>
150      <cell_definition name="bacteria" ID="0">
151        <phenotype>
152          <cycle code="5" name="live">▭</cycle>
157          <death>▭</death>
188          <volume>▭</volume>
199          <mechanics>▭</mechanics>
219          <motility>▭</motility>
246          <secretion>▭</secretion>
278          <cell_interactions>▭</cell_interactions>
309          <cell_transformations>▭</cell_transformations>
320        </phenotype>
321        <custom_data>▭</custom_data>
```

**Figure 1.** Portion of the XML configuration file showing elements of a cell definition.

with every PhysiCell download. The "template" model defines one cell type ("default") and one substrate ("substrate"). In this model, a specific (predefined) cell cycle is defined that results in proliferation and the cell death parameters result in apoptosis. The other cell phenotype parameters use defaults provided by PhysiCell: "standard" mechanics, no motility, no secretion, etc. The substrate has an identically zero initial condition and a zero-flux (Neumann) boundary condition with no additional Dirichlet boundary conditions. There is a user parameter, "number_of_cells", that defines the number of initial cells positioned randomly and uniformly in the spatial domain.

We start by copying this template (.xml) model into a new file, tumor_demo.xml. There are a few different workflows for using the Studio. However, the most common way is discussed in the Studio Guide [9]. Assuming the Studio is installed in a PhysiCell root directory and you have compiled the template project executable (called "project"), then you can create the new tumor_demo.xml and run the Studio from the command line with the following (adjusting the syntax for Windows if necessary, rf. Studio Guide):

```
~/PhysiCell$ cp studio/config/template.xml   tumor_demo.xml
~/PhysiCell$ python studio/bin/studio.py  -c tumor_demo.xml  -e project
```

(If you need help installing PhysiCell, in order to build the template project, see [10].)

You should see something similar to Figure 2 that displays the first tab for Configuration Basics parameters for your model (same as the template model). In this tab, you can configure the size of the domain, set time stepping parameters, set the frequency of collecting model data, and more [11]. For now, check the "enable" checkbox in the "Initial conditions of cells" section as we will be using it for the tumor model.

Next, select the "Microenvironment" tab, where the substrates (or signals) are defined. In the template model, you will see a "substrate" defined. This tab will display all substrates in the model in the left panel along with buttons to create new substrates, copy substrates, and delete substrates. On the right, you will see all the available parameters for the highlighted substrate on the left. In addition, two checkboxes appear at the bottom that change the behavior of the model for all substrates: "calculate gradients" and "track in agents".

Perform the following steps to set up oxygen in the model: (i) select "substrate" – double-click the name – and rename it "oxygen"; (ii) set the "decay rate" to 0.1 (to give a 1 mm $O_2$ diffusion length scale outside the tumor); (iii) set the "initial condition" to 38

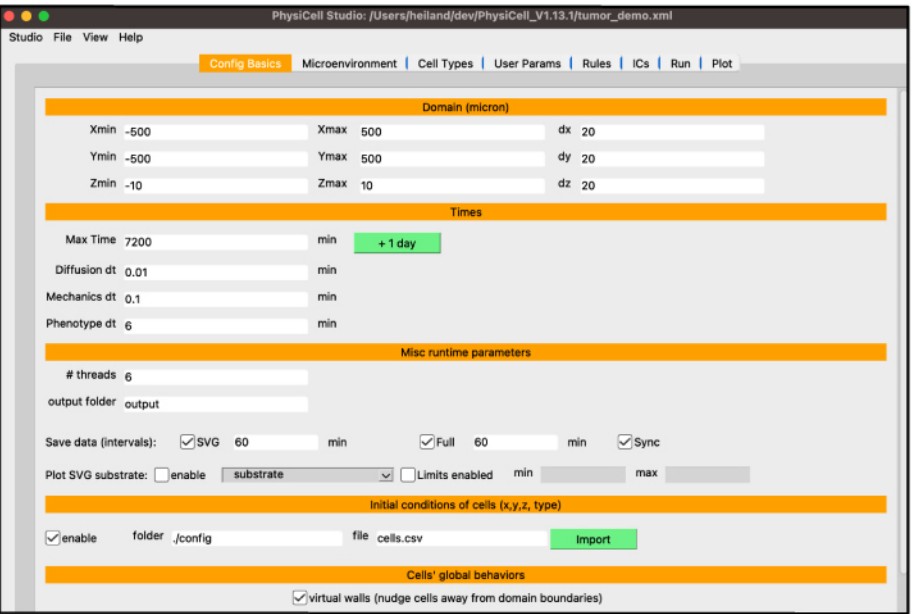

**Figure 2.** View of the basic configuration of a PhysiCell model in PhysiCell Studio.

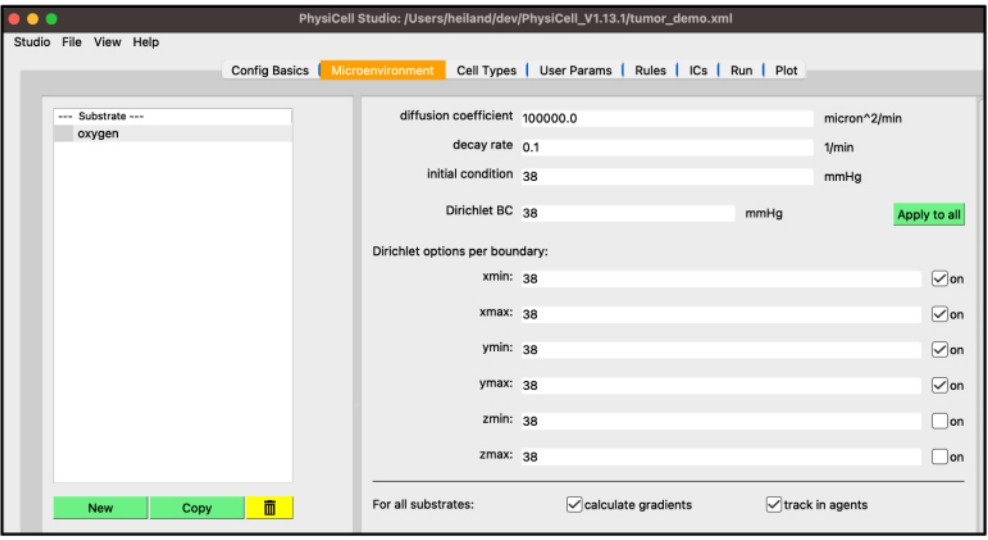

**Figure 3.** View of the microenvironment configuration of a PhysiCell model in PhysiCell Studio.

(a 5% physioxic condition); (iv) set the "Dirichlet BC" – Boundary Condition – to 38; and (v) press "Apply to all" then check the "on" checkbox for xmin, xmax, ymin, and ymax.

Your screen should look like Figure 3.

Next, select the "Cell Types" tab. True to its name, PhysiCell is an agent-based model of cells, hence the most detail goes into defining the cell types. That is why this tab contains the most information, organized by nine subtabs. Similar to the "Microenvironment" tab, the left panel shows the list of current cell types as well as their ID, a non-negative integer

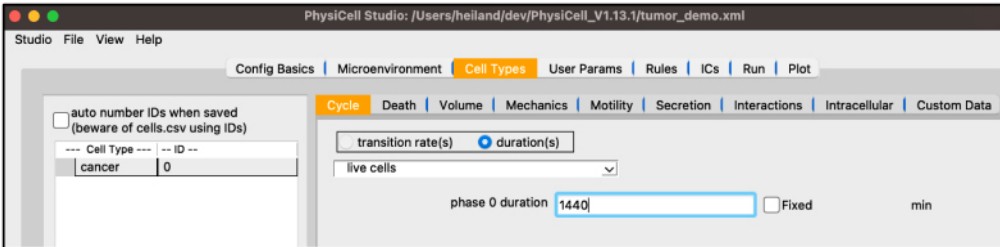

**Figure 4.** View of the cell cycle definition of a cancer cell in PhysiCell Studio.

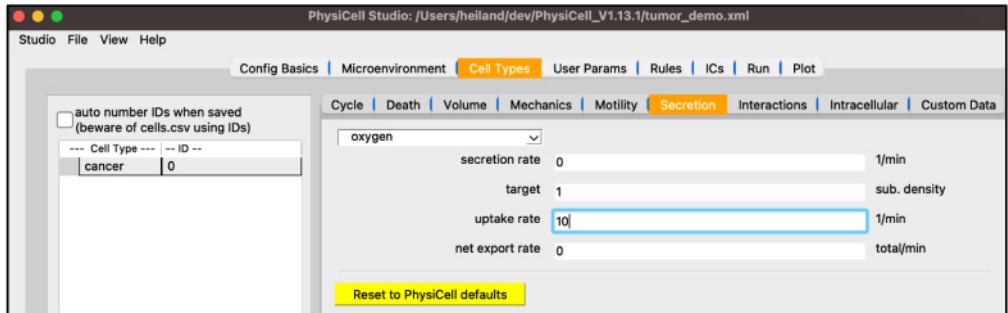

**Figure 5.** View of the secretion configuration of a cancer cell in PhysiCell Studio.

that you only need to account for if you create your own custom C++ code referring directly to cell type IDs. (See the PhysiCell biorobots sample project: "robot_coloring_function" in custom_modules/custom.cpp.) On the right, you can cycle through the nine subtabs, displaying the related information for the highlighted cell type on the left. In the template model, you will see a "default" cell type defined.

Select "default", double-click the name, and rename it "cancer". The "Cycle" tab should already be selected, but if not, select it. Click the dropdown widget containing predefined cell cycles and select "live cells" (a simpler cell cycle representation [12, 13]). Be sure the "duration(s)" radio button is selected and set the "phase 0 duration" to 1440 (min, i.e., 24 h).

Your screen should look like Figure 4.

Staying in the "Cell Types" top tab, select the "Secretion" subtab. Note its dropdown widget only lists "oxygen" since that's the only substrate defined so far. In a model with more substrates, those will automatically be added to this dropdown for you to select and update the four cell-type-specific parameters shown below.

Set the "uptake rate" to 10. (This corresponds to a 100-micron length scale.)

Your screen should look like Figure 5.

Next, we will create the initial conditions (i.e., the initial cell positions) for the circular tumor. Select the "ICs" tab. In this tab, initial cell locations can be set using a graphical interface. Select the cell type from the dropdown widget, use the two dropdown widgets below to set how you will add cells, fine tune your placement with parameters, and "Plot" the result. If you choose "point" from the first dropdown, you can click on the figure in the right panel to add cells at specific locations.

The top "Cell Type" dropdown widget should only contain "cancer". Be sure "annulus/disk" is selected in the geometry dropdown. Select "hex fill" in the fill options

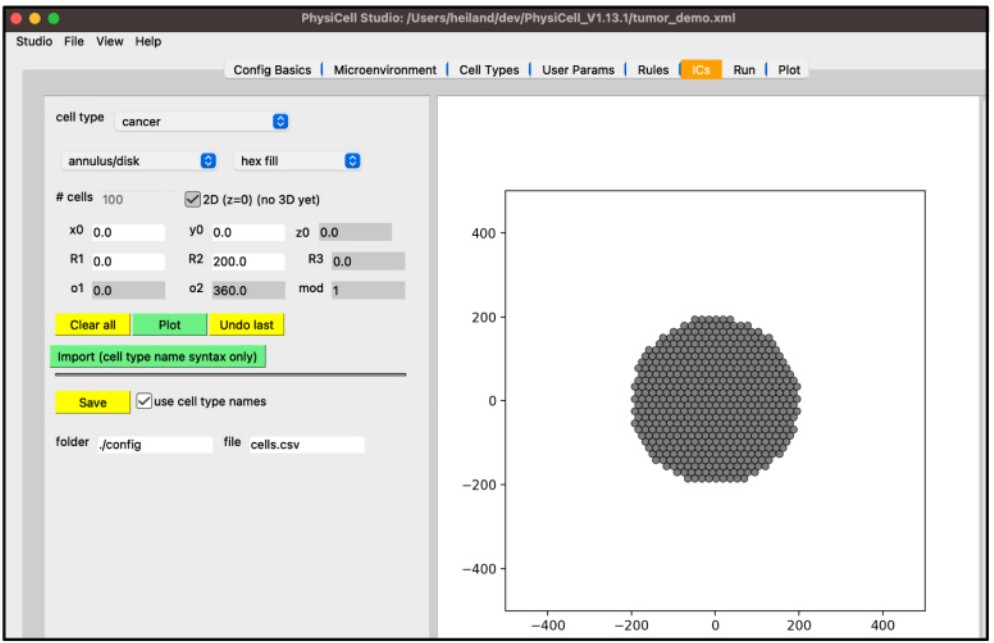

**Figure 6.** View of the definition of cells' initial conditions of a PhysiCell model in PhysiCell Studio.

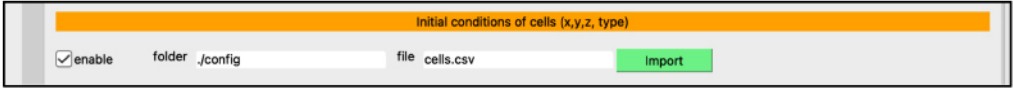

**Figure 7.** Enable cells' initial conditions on the Config Basics tab.

dropdown. Set R1 (minimum radius) to 0. Set R2 (maximum radius) to 200. Click "Plot". Click "Save".

Your screen should look like Figure 6. After any change to these initial conditions, you must click "Save". PhysiCell Studio only saves to the CSV when this button is pressed, not when you "File → Save" the XML.

In the "Config Basics" tab, confirm that you have checked "enable" for the initial conditions (Figure 7). If you do not, these initial conditions will not be loaded into your simulation.

Next, select the "User Params" tab. You will see a table of user parameters that you can add to, modify, or delete from. The first three columns of this table are required by PhysiCell while the final two are for interpretability. Only those user parameters that display upon initial loading of Studio with the sample project – those that the sample project's C++ uses – will affect the simulation. Adding additional user parameters in the Studio would only make sense if the model's C++ code uses them.

Set the "number_of_cells" to 0 (so that we only have our hex-packed disk of cells). This user parameter sets the number of randomly positioned agents (of each cell type) in the "template" sample project; this is only needed when cell positions aren't explicitly supplied as an initial condition as in our example.

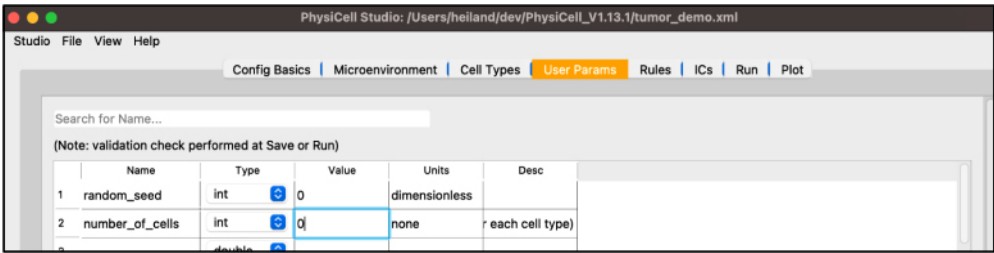

**Figure 8.** View of the definition of user parameters in PhysiCell Studio.

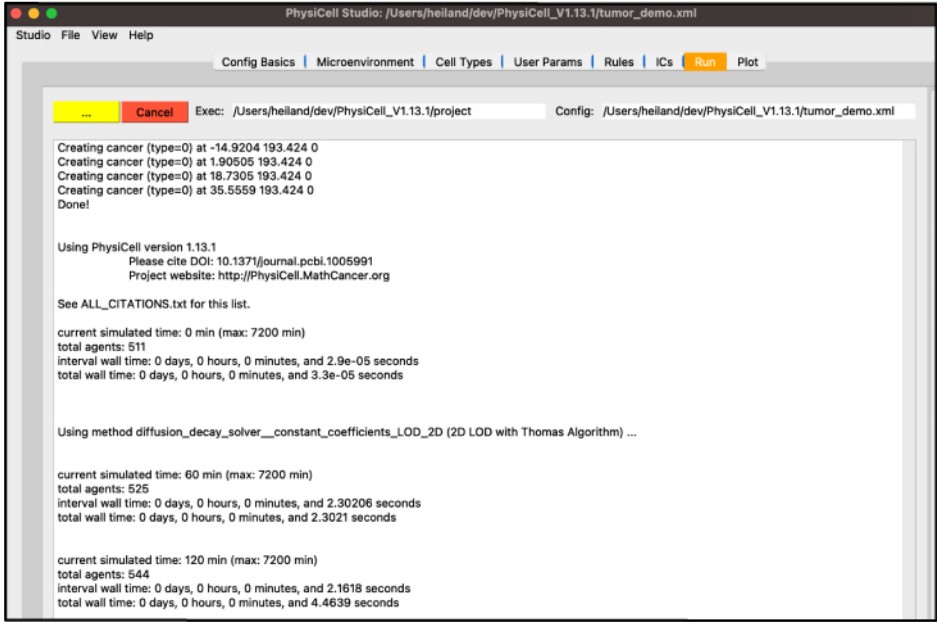

**Figure 9.** View of the simulation log in PhysiCell Studio.

Your screen should look like Figure 8.

Next, select the "Run" tab and click "Run simulation". This will cause all edits you have made so far to be saved into "tumor_demo.xml". Additionally, PhysiCell Studio uses the inputs you gave to launch it to populate the executable and configuration files for you. The simulation will run, showing the normal terminal output in this tab (Figure 9).

While the simulation is running, navigate to the "Plot" tab. In this tab, you can advance through snapshots of the model as it is running by navigating with the arrows, entering a specific snapshot ID, or clicking "Play" and watching a movie of the recorded output. For the best experience, select the "Sync" option on "Config Basics" in the "Save data (intervals)" row to synchronize the cellular and substrate snapshots. Many options exist for what data to display, including cell-specific data (pressure, cycle phase, etc.) and individual substrate concentrations. By clicking the "Legend (.svg)" button, a legend will appear in a new window identifying the cell types. Clicking the "Population plot" will open a new window with time series corresponding to the item in the dropdown widget.



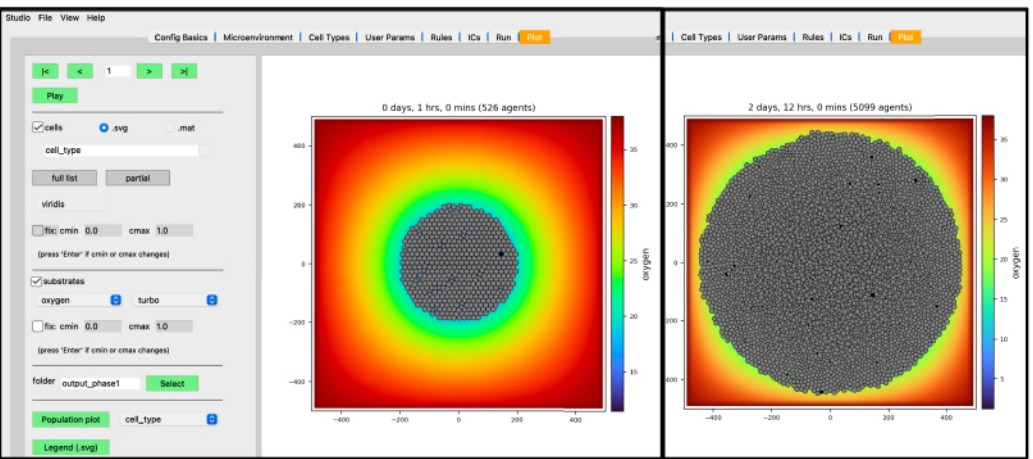

**Figure 10.** Simulation results at 1 h (left) and 2 days, 12 h (right) in PhysiCell Studio.

For now: (i) leave the "cells" checkbox checked and the ".svg" radio button selected; (ii) check the "substrates" checkbox to plot the diffusing oxygen substrate and choose "turbo" in the colormap dropdown; and (iii) press the " >" button to advance a single frame.

Your screen should look similar to Figure 10 (left). If you press the "Play" button, it should animate results from the simulation. Figure 10 (right) shows results at 2.5 days. Be aware that results from PhysiCell simulations will be stochastic if you are using more than one OpenMP thread, so there will be some variability between runs.

We have modeled a growing tumor whose cells uptake oxygen. One thing to note is the tumor cells overlap in a non-realistic manner. This can be made more obvious if we plot the tumor cells color-coded by how much pressure is exerted on each one (Figure 11).

To correct this non-realistic outcome, we can define a pressure mechano-feedback rule. A rule defines a cell behavior as a function of some signal, providing a powerful modeling feature of PhysiCell [14]. This, along with more extensions to this tumor model, can be found in the archived 'Supplementary data' in Zenodo [15]. (We also provide a living version of the 'Supplementary data' at [16].)

## DESIGN AND DEVELOPMENT

PhysiCell Studio has been designed and developed by academic researchers. Graduate students, as early users, have provided valuable feedback and contributions. In our lab at Indiana University, we also include undergraduate students in research projects, and this has definitely been true for the Studio. By combining graduate and undergraduate students in regular lab meetings, we foster both education and research. Undergraduates learn more about active research projects (like the Studio and models developed using it); graduate students, postdocs, and staff become mentors [17].

One key design goal was to have PhysiCell Studio be an independent project from PhysiCell. By "independent" we mean that, first, it should not affect the legacy workflow for using PhysiCell. A modeler should still be able to edit the XML model by hand, run a simulation from the command line, and visualize output results however they wish. Second, we want the Studio to have an independent development path, likely with more frequent software releases than PhysiCell. However, the latest version of the Studio should always

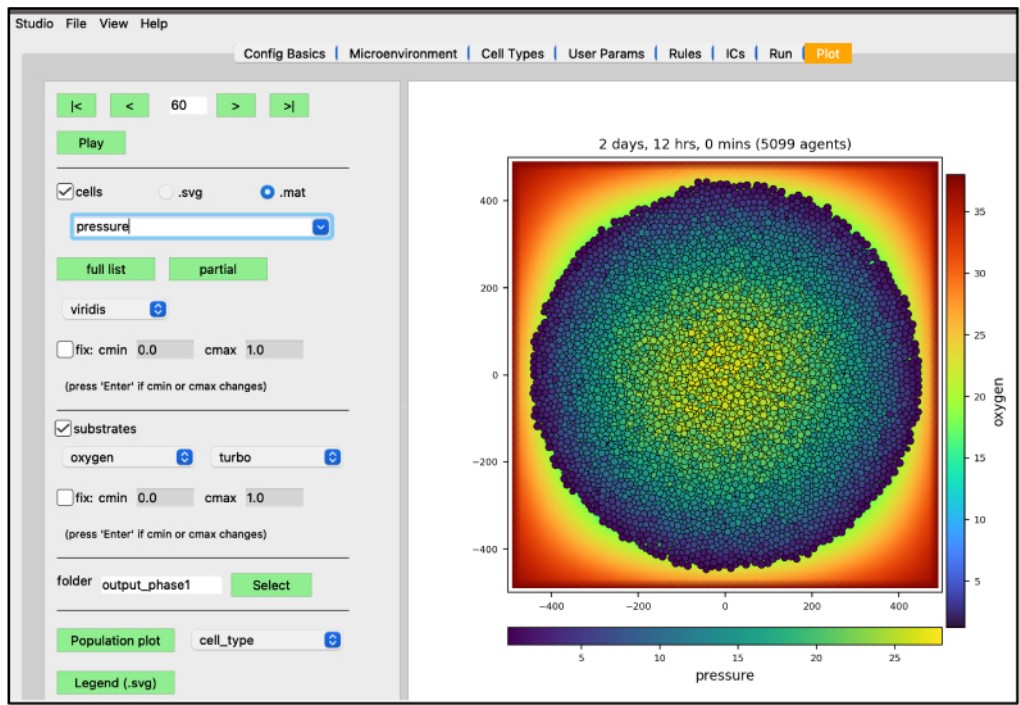

**Figure 11.** Pressure values on cells (at 2 days, 12 h) in PhysiCell Studio.

expose whatever model parameters are available in the latest version of PhysiCell. And third, it allows for having different software licenses.

The development of PhysiCell Studio has progressed in stages. In the first stage, as part of a National Science Foundation nanoBIO grant [18], we developed a Python (RRID:SCR_008394) script and workflow to transform an existing PhysiCell model configuration file (XML) into a Jupyter notebook with a GUI (Figure 12). Undergraduate students played an active role during this stage and contributed to the xml2jupyter project [19]. When the GUI was combined with the PhysiCell C++ code base and custom C++ for that particular model, the model was accessible from a Web browser, parameter values could be modified, and a simulation executed in the cloud on the nanoHUB platform (e.g., nanohub.org/tools/pc4covid19). In addition, 2D simulation results could be visualized in the same tool. The ability to edit the model, however, was limited to modifying values of existing parameters. A user could *not* add more (nor delete, nor rename existing) objects or parameters in the model using the GUI. Nevertheless, the layout of the GUI during this stage influenced the layout of PhysiCell Studio.

The second stage of development was to prototype a desktop tool that resembled the Jupyter notebook's layout and functionality but was more powerful. It needed to be able to: (1) add (or delete or rename) objects such as substrates, cell types, custom data parameters, and user parameters; and (2) define associations between objects, such as cell type C {secretes, or chemotaxis to/from} substrate S, or cell type C1 {interacts with} cell type C2. All model edits performed in the Studio would then be saved in the XML configuration file.

We chose Qt [20] as the preferred GUI library for the desktop tool for multiple reasons. It is used by several other desktop (scientific) applications, runs on the three major operating

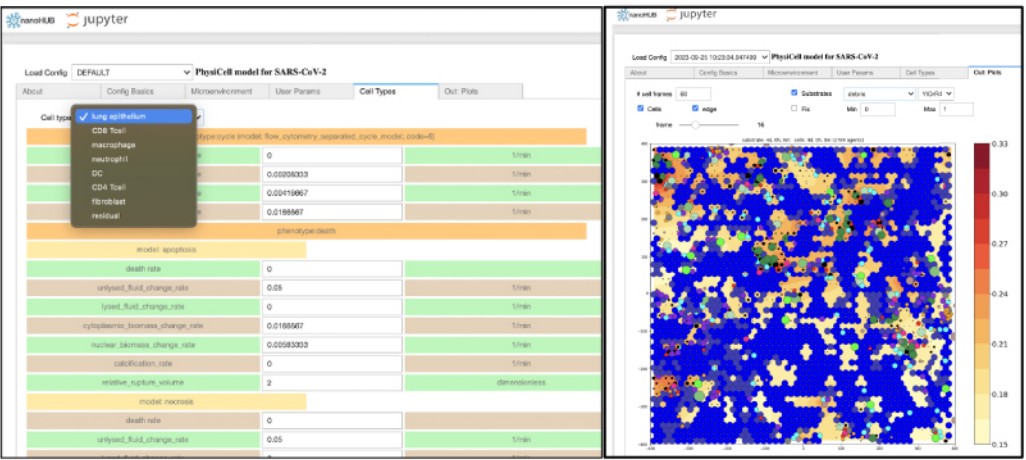

**Figure 12.** Jupyter notebook GUI showing some model parameters (left) and simulation results (right).

systems (Windows, macOS, and Linux) that PhysiCell supports, and has Python Application Programming Interfaces (APIs). Undergraduate students explored the C++ API to Qt, along with Qt Designer [21] to create a primitive prototype of the desktop Studio. Others explored a Python API to Qt [22] to do the same. In the end, we selected the Python API for two reasons: (1) the development cycle was faster (no compilation required); and (2) we believed the PhysiCell community (who might further develop the Studio) would be more familiar with Python than C++. (This was also true for undergraduate and graduate students.)

We made early prototypes of the Studio available to lab members, close collaborators, and community leaders who attended PhysiCell workshops. Their feedback led to a third stage of development that included the ability to create cells' initial conditions (in 2D) and additional visualization functionality, including cells' scalar values (in the .mat output files) and plotting for 3D models (using VTK) [23].

Finally, a fourth stage of development provided a graphical interface to a recent, powerful modeling concept in PhysiCell: cell behaviors can be interactively defined as responses to signals (stimuli) [14]. These behaviors are specified using a constrained grammar, leading to model "rules" (Figure 13). As a modeler adds (or deletes or renames) substrates or cell types, in their respective tabs, the widgets in the "Rules" tab for signals and behaviors will be dynamically updated, in addition to any rules already defined and listed in the table.

One general usability feature of the Studio is worth mentioning. It operates mostly in "immediate mode"; confirmation of an action is not required. For example, in the Plot tab, clicking on a widget or changing a text value will, most of the time, cause an immediate, visible change in the plot window. One exception is the "cmin" or "cmax" value that pertains to the colorbars. If a user changes either one of these values, they need to press the "Enter" key ("Return" on Mac) for the plot results to be updated. There is text next to those widgets as a reminder. The reason for this required action is because it may be an expensive operation. In other tabs, for example the Cell Types, entering a new value in a text parameter widget does not require pressing "Enter" for it to be saved (to intermediate data structures). A related design feature is that we store all XML objects and parameter values in internal Python dictionaries (the intermediate data structures) during a Studio session.



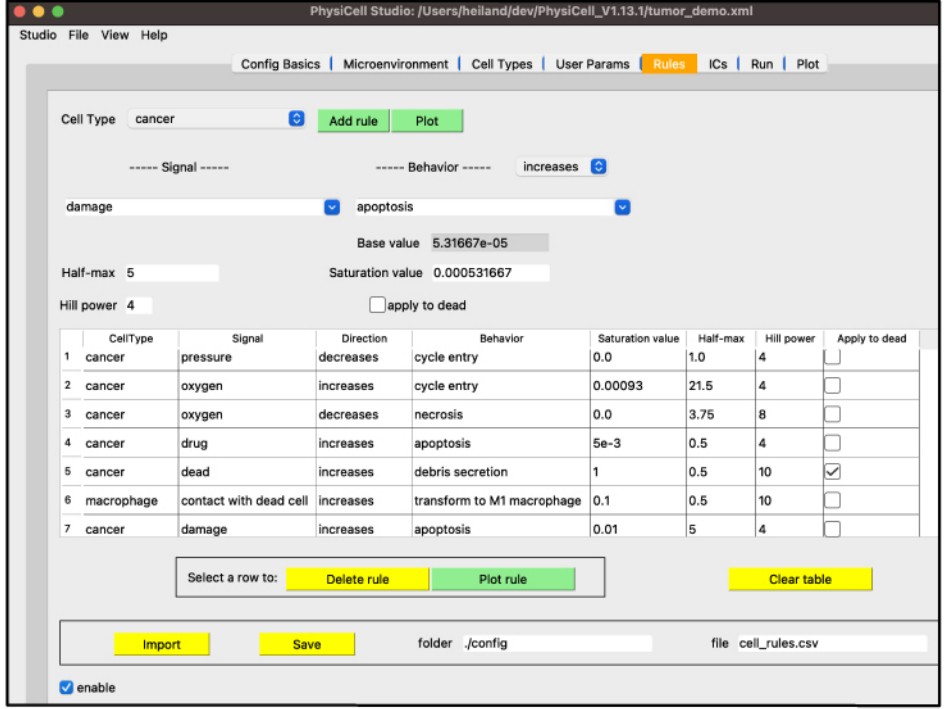

**Figure 13.** An example of model rules (see 'Supplementary data') in PhysiCell Studio.

The contents of these dictionaries will be written to an XML file when the user explicitly does a "File → Save" (or "Save as") or performs a "Run Simulation" in the Run tab.

## GRAPHICS: PLOTTING AND INITIAL CONDITIONS

During the early design of PhysiCell Studio, we had considered creating it as just a model editor, with no plotting functionality. There were plenty of scientific visualization libraries and tools, both commercial and open source, that a modeler could use to post-process results of a PhysiCell simulation, such as MATLAB (RRID:SCR_001622), matplotlib [24], VTK [23], ParaView, and Simularium [25]. We decided it was worth the effort to include some degree of interactive visualization within the Studio, offering benefits such as: (1) avoiding a potential steep learning curve using other plotting tools; (2) avoiding cognitive context switching between tools; and (3) reducing the time to develop a model (the edit→run→visualize cycle).

PhysiCell Studio uses the matplotlib library (RRID:SCR_008624) [24] for 2D and VTK (its Python API, see vtk.org) for 3D visualizations [23] (RRID:SCR_015013). However, a very limited and targeted subset of functionality from those libraries is used and exposed in the GUI. Nevertheless, there are plenty of challenges when visualizing any scientific data. For PhysiCell data, we need to interactively plot possibly hundreds of thousands of cells, changing position, size, and color (where color is specified in either SVG or scalar values). In addition, there may be multiple scalar fields representing the microenvironment, such as oxygen, glucose, chemokine, and interferon, that also need to be interactively rendered. Therefore, the Studio offers choices for colormaps and an option to clamp its scalar range.

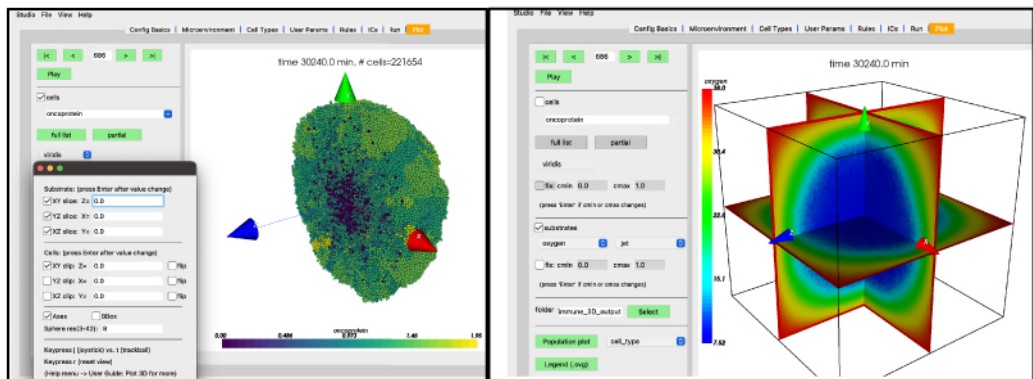

**Figure 14.** 3D plots of the cancer-immune-sample model in PhysiCell Studio. Left: Clipping plane (XY) of a cell population. Right: Slice planes (XY, YZ, XZ) showing projected substrate concentrations.

There are additional challenges with 3D models and data, requiring, for example, the need to hide (clip) data or extract 2D subsets (Figure 14). See [26] for more details. The 3D visualization available in the current version should be considered early prototyping and will be improved in future versions. The Studio will never meet everyone's needs for visualizing simulation data, but we try to provide a sufficient set of options and can expand it when the community generally agrees they need more.

In addition to plotting output data from a simulation, the Studio can also generate input data (currently 2D only). Specifically, the "ICs" tab lets a user graphically create initial conditions for cells. (In the future it will also provide ways to create initial conditions for substrates.) By selecting a cell type, a geometric region, the type of fill (random or hexagonal), plus additional parameters, one can generate a .csv file for cells' initial conditions. Figure 15 shows a circular region of tumor cells and an outer annulus of immune cells.

## INTRACELLULAR MODELING

PhysiCell Studio will support intracellular modeling. Currently only a boolean intracellular modeling interface is provided for the PhysiBoSS [27] add-on, allowing settings edits and specific visualization. See Figure 16. In the future, we will also provide an interface for ODE intracellular models [28] and be receptive to others the community may want.

## SOFTWARE ENGINEERING

We have adopted a software engineering workflow that uses GitHub [29] and takes advantage of several features it offers: hosting and distributing software, discussing issues, submitting pull requests, developing code in staged repositories, and automated testing.

Most of our community is already familiar with GitHub, but for those who are not, we help them learn the basics. For anyone who wants to contribute code to the Studio, we ask that they fork the repository into their own account, make edits, test (on at least one of the three supported operating systems), and make pull requests to the development branch of the Studio repository. Community discussion about bugs (and hopefully proposed solutions) or new features for the Studio is encouraged via Slack channels and GitHub Issues. (See Community Support below.)

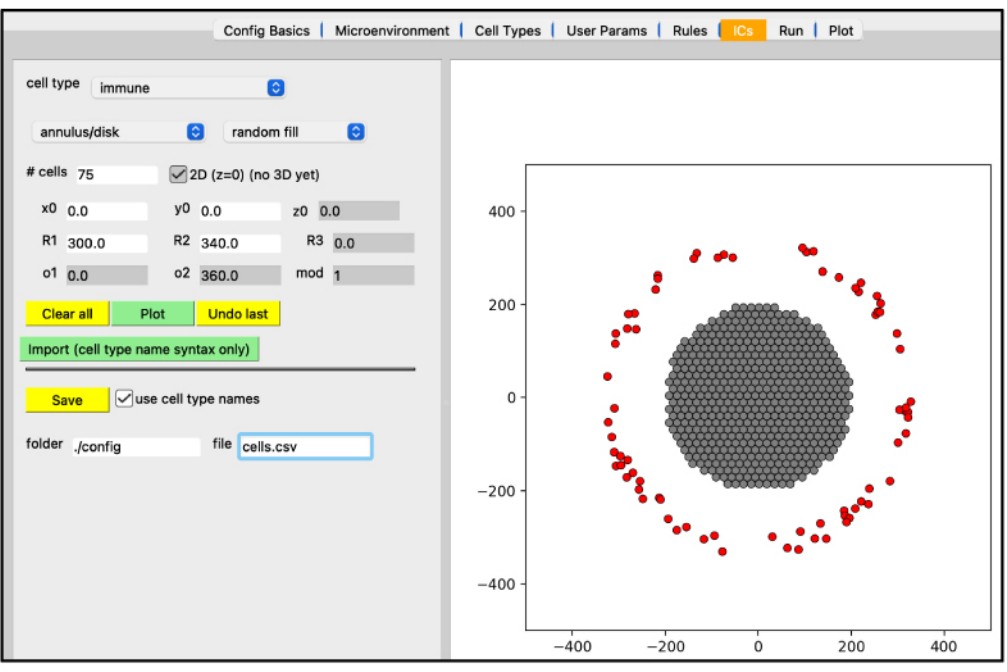

**Figure 15.** Creating initial conditions for cells.

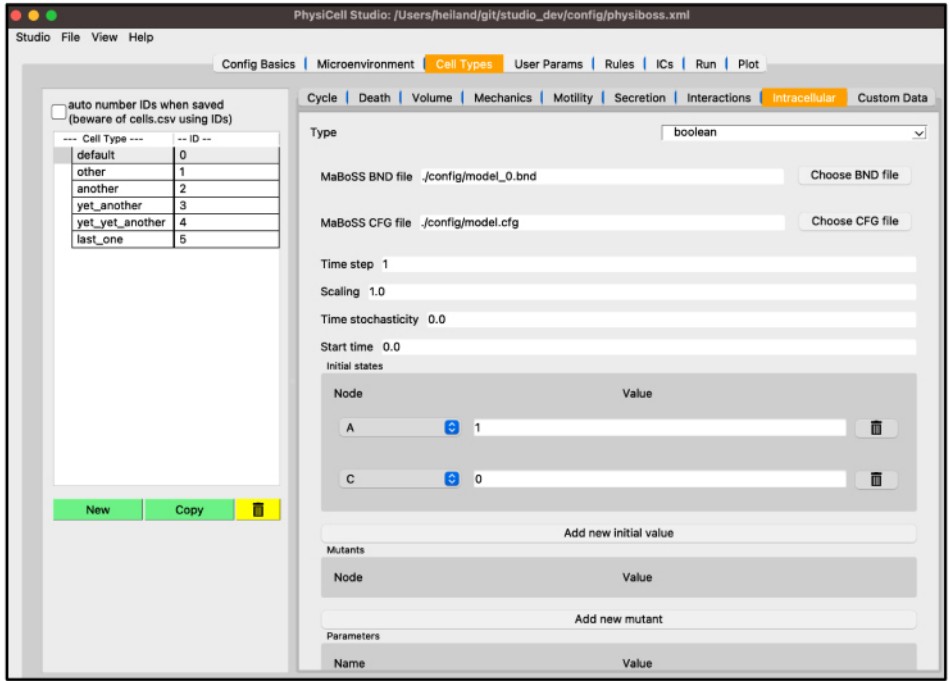

**Figure 16.** Interface for a boolean intracellular model in PhysiCell Studio.



The Studio uses Python logging to capture significant actions that occur during a modeling session. This log file can then be shared with developers in the event of a fatal error or unusual behavior.

For automated testing, we use pytest, a very popular tool for Python applications, and pytest-qt, a pytest plugin for testing Python APIs to Qt applications [30]. We have only recently begun automated tests but will be adding more as the Studio evolves. Not only is there a large parameter space in a PhysiCell model, there is also a large "parameter" space (user-widget actions) that can occur in a Studio session. Automated tests are necessary to ensure ongoing development of the software does not introduce unwanted results.

## BUNDLING AND DISTRIBUTING

The current version of the Studio does not bundle any pre-built PhysiCell executable model, pre-built library, or C++ code in its distribution. Therefore, a user will still need to download and build an executable model which can then be used in the Studio's "Run" tab to run a simulation. In the future, we will likely provide bundled distributions of the Studio which will include both a minimal Python distribution and a "template" PhysiCell executable model.

Since the Studio uses a Python API to Qt, Python is one dependency. Python's standard library provides many useful data structures and an efficient XML API module for handling much of the functionality in the Studio. However, it will also need modules that are not in the standard library: PyQt5 (GUI) [22], matplotlib [24] and numpy (2D plotting, numerical computing) [31], scipy (reading .mat files) [32], and VTK (3D plotting) [23]. For PhysiCell (and Studio)-related workshops and university courses, we typically ask users to install the free Anaconda Python distribution [33]. Although it is relatively large and provides many more modules than the Studio needs, experience has shown that users will avoid many potential problems by using it. In addition, some of those extra Python modules may later prove to be useful – for example, doing data analysis on PhysiCell output results. However, we also provide setup files for a more limited set of dependencies if users want to try that approach [34].

## COMMUNITY SUPPORT

The PhysiCell Studio User Guide [35] should help new users get started. For additional support, see [36]. An introductory video from a recent PhysiCell workshop is available to view (Figure 17, [37]). More details about defining Rules using the constrained grammar for cell behaviors can be found in that paper's supplementary material [14]. We are always open to new ideas for learning how to use PhysiCell Studio and welcome community contributions.

## INTERFACING TO OTHER TOOLS

PhysiCell Studio will never provide everything that users need. There will always be additional functionality that modelers want, whether it be something mundane such as creating a montage of output images for a publication, something computationally intensive like data analysis on a model's parameter space exploration, or numerous other things. To help bridge the gap to other tools, we provide functionality that transforms output data into other formats. In collaboration with a team at the Allen Institute for Cell Science, the Studio can generate data (File → Export → Simularium) for their Simularium [38] viewer (simularium.allencell.org/viewer) shown in Figure 18.

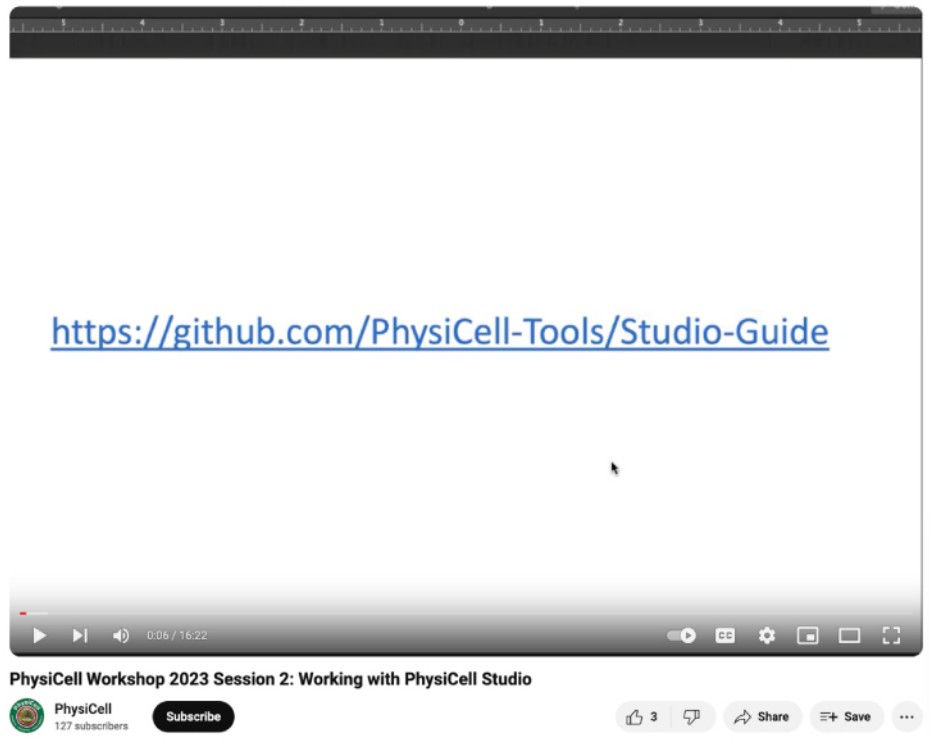

**Figure 17.** Working with PhysiCell Studio. A video from Session 2 of the PhysiCell Workshop 2023 [37]. https://youtu.be/jkbPP1yDzME.

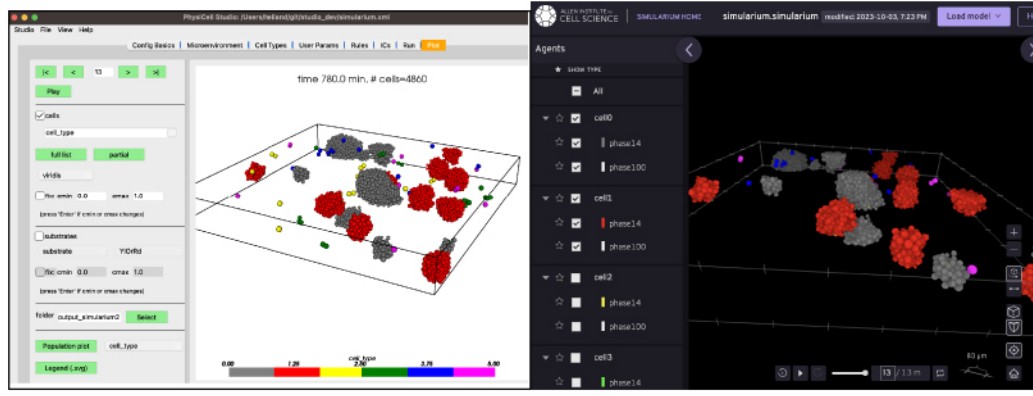

**Figure 18.** Studio 3D display (left) and the Simularium viewer (running in a web browser) which allows cell types to be hidden (right).

ParaView (RRID:SCR_002516, paraview.org) is a very popular open-source desktop tool for scientific visualization. There is no direct interface from PhysiCell Studio to ParaView, but we provide a customized workflow that lets ParaView render output data from a PhysiCell simulation. This workflow, along with the necessary Python scripts and example ParaView state files are provided at [39]. An example is shown in Figure 19.

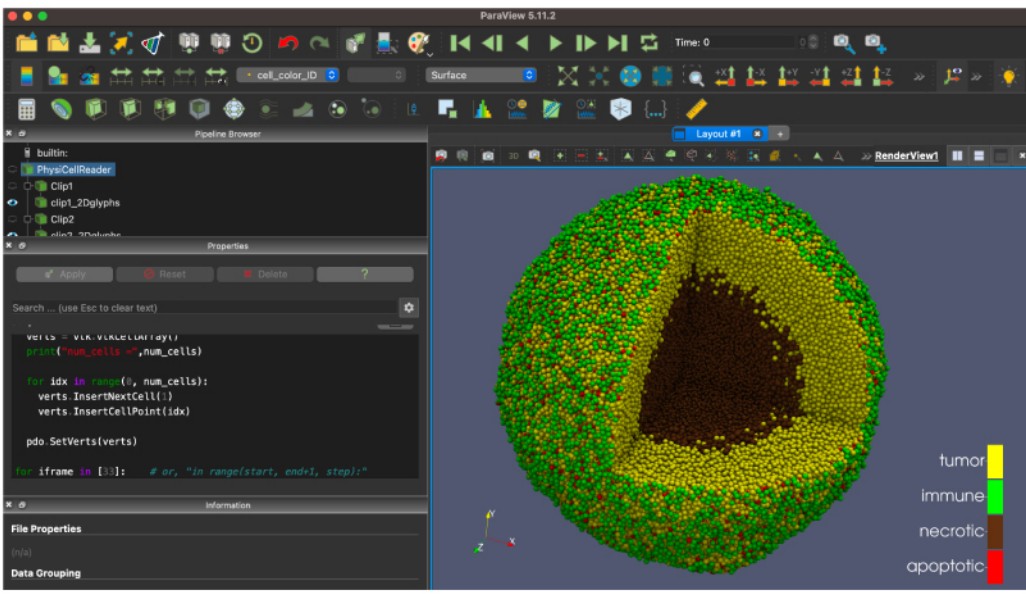

**Figure 19.** ParaView rendering of data from the cancer-immune-sample model in PhysiCell Studio.

The trade-offs of providing functionality in the Studio versus using other tools, especially for visualization, is an ongoing challenge and we strive to maintain a balance. But the community will need to provide feedback and contributions for additional data format transformations for other tools.

## DISCUSSION

Developing PhysiCell Studio has been a somewhat lengthy, iterative process in an academic environment where multiple projects required our attention. Developing any GUI has unique challenges. For the Studio, it tries to: (1) help a user create and maintain a mental model of interacting objects in a multicellular system (e.g., cells with phenotypic behaviors and signals in the microenvironment); and (2) manage user expectations of GUI actions (e.g., clicking a button or selecting an item in a dropdown widget). Although we had had some past experience developing GUIs for computational science [40–43], we had lacked formal training in human–computer interaction (HCI) – an entire academic field. We had also lacked formal user studies during the development of PhysiCell Studio, beyond our extensive testing in day-to-day, real-world scientific workflows. In spite of these shortcomings, we believe the end result is an extremely useful tool which seems to be quite popular, both with seasoned PhysiCell modelers and with new users just learning PhysiCell.

This paper has presented the desktop tool version of PhysiCell Studio (version 2.34.3, and PhysiCell 1.13.1). In addition, we provide an interactive version that runs in a web browser at nanohub.org/tools/pcstudio. (Access requires creating a free nanoHUB account [44].) Unfortunately, the browser version currently lags behind the desktop version, so there will be slight differences in the GUI and the functionality. We plan to synchronize their code bases in the future. In addition, there are interactive PhysiCell training modules that can be run in the browser [45].

Looking to the future, we are planning to add new features based on community feedback and contributions. In terms of promoting even broader accessibility, it would be interesting to explore the Qt speech interface [46] at some point.

## SUMMARY

We have presented PhysiCell Studio, an open-source desktop tool that provides a GUI for building, simulating, and visualizing a PhysiCell model. The Studio has gone through several iterations of development and benefited from user feedback at several PhysiCell workshops and university classes. The end result is a transformative tool for developing a multicellular model, not only for new users, but also for experienced PhysiCell modelers. The process of designing and developing the Studio has involved both graduate and undergraduate students, as well as several members in the larger PhysiCell community.

## AVAILABILITY OF SOURCE CODE AND REQUIREMENTS

- Project name: PhysiCell Studio
- Project home page: https://github.com/PhysiCell-Tools/PhysiCell-Studio
- Operating system(s): Platform independent
- Programming language: Python
- Other requirements: Six additional Python modules (matplotlib, VTK, numpy, scipy, PyQt5, and anndata) not included in the Python standard library, but freely available and supported.
- License: GNU GPL v3
- RRID:SCR_025311.

## DATA AVAILABILITY

Supplementary data (including a detailed continuation of the tumor model walk-thru) are available in Zenodo [15]. Archives of the software are available in Software Heritage [47].

## ABBREVIATIONS

API, Application Programming Interface; GUI, Graphical User Interface; SVG, Scalable Vector Graphics; XML, eXtensible Markup Language.

## DECLARATIONS

### Ethics approval and consent to participate

The authors declare that ethical approval was not required for this type of research.

### Competing interests

The authors declare they have no competing interests.

### Authors' contributions

Conceptualization: RH, PM; Writing original draft: RH; Writing - review and editing: RH, DB, BL, GW, JC, HLR, MR, VN, PM; Software: RH, DB, BL, GW, JC, HLR, MR, VN, PM; Funding acquisition: VN, PM.

### Funding

We thank the National Science Foundation (Awards 1720625 and 2303695), the National Institutes of Health (U01-CA232137-01), and the Jayne Koskinas Ted Giovanis Foundation for



Health and Policy. This work was also supported by the European Commission under the PerMedCoE project (H2020-ICT-951773) and the Inserm amorçage project.

## Acknowledgements

We thank the entire PhysiCell community for providing helpful feedback and contributions to the Studio, including several undergraduate students over the past few years: Adam Morrow, Daniel Mishler, Tyler Zhang, Eric Bower, Carlos Juarez, Jay Thilking, Nicholas Goh, Yuchen Yang, Drew Willis, Kimberly Crèvecoeur, Dylan Taylor, Kali Konstantinopoulos, Marshal Gress, and Eric Freeman, as well as graduate students: John Metzcar, Elmar Bucher, Furkan Kurtoglu, Aneequa Sundus, Yafei Wang, Supriya Bidanta, and postdoc Michael Getz. We also thank Steven Clark, Daniel Mejia, Martin Hunt, and Lynn Zentner for their support with nanoHUB. Finally, we thank many open-source software communities for their support: Python, matplotlib, ParaView, VTK, and more.

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
