## [Editor Report]

Editor’s AssessmentThis paper presents a new tool to make using PhysiCell easier, which is an open-source, physics-based multicellular simulation framework with a very wide user base. PhysiCell Studio is a graphical tool that makes it easier to build, run, and visualize PhysiCell models. Over time, it has evolved from being a GUI to include many additional functionalities, and can be used as desktop and cloud versions. This paper outlines the many features and functions, the design and development process behind it, and deployment instructions. Peer review improved the organisation of the various repositories and adding both a requirements.txt and environment.yml files. Looking to the future the developers are planning to add new features based on community feedback and contributions, and this paper presents the many code repositories if readers wish to contribute to the development process.Editor’s AssessmentThis paper presents a new tool to make using PhysiCell easier, which is an open-source, physics-based multicellular simulation framework with a very wide user base. PhysiCell Studio is a graphical tool that makes it easier to build, run, and visualize PhysiCell models. Over time, it has evolved from being a GUI to include many additional functionalities, and can be used as desktop and cloud versions. This paper outlines the many features and functions, the design and development process behind it, and deployment instructions. Peer review improved the organisation of the various repositories and adding both a requirements.txt and environment.yml files. Looking to the future the developers are planning to add new features based on community feedback and contributions, and this paper presents the many code repositories if readers wish to contribute to the development process.

---

## [Reviewer Report]

Upload additional filesTRR-202401-01R01/stage_files/TRR-202401-01/Review MS/Review_PhysiCellStudio_MV.pdfReviewer name and names of any other individual's who aided in reviewerMeghna VermaDo you understand and agree to our policy of having open and named reviews, and having your review included with the published manuscript. (If no, please inform the editor that you cannot review this manuscript.)YesIs the language of sufficient quality?YesPlease add additional comments on language quality to clarify if neededIs there a clear statement of need explaining what problems the software is designed to solve and who the target audience is? YesAdditional CommentsIs the source code available, and has an appropriate Open Source Initiative license <a href="https://opensource.org/licenses" target="_blank">(https://opensource.org/licenses)</a> been assigned to the code?YesAdditional CommentsAs Open Source Software are there guidelines on how to contribute, report issues or seek support on the code?YesAdditional CommentsIs the code executable?YesAdditional CommentsIs installation/deployment sufficiently outlined in the paper and documentation, and does it proceed as outlined?YesAdditional CommentsThe authors have provided links for video descriptions for installation and that is appreciated. Is the documentation provided clear and user friendly?YesAdditional CommentsIs there enough clear information in the documentation to install, run and test this tool, including information on where to seek help if required?YesAdditional CommentsIs there a clearly-stated list of dependencies, and is the core functionality of the software documented to a satisfactory level?YesAdditional CommentsHave any claims of performance been sufficiently tested and compared to other commonly-used packages? Not applicableAdditional CommentsAdditional CommentsAre there (ideally real world) examples demonstrating use of the software? YesAdditional CommentsAdditional CommentsAny Additional Overall Comments to the AuthorOne overall recommendation is: If all the screenshots (for e.g.: from Fig 1-12 of the main paper and all the subsections in Supplementary) can be combined in one figure that will help enhance the complete overview and the overall flow of the paper.RecommendationMinor Revisions

---

## [Reviewer Report]

Reviewer name and names of any other individual's who aided in reviewerKoert Schreurs and Lin Wouters supervised by Inge WortelDo you understand and agree to our policy of having open and named reviews, and having your review included with the published manuscript. (If no, please inform the editor that you cannot review this manuscript.)YesIs the language of sufficient quality?YesPlease add additional comments on language quality to clarify if neededIs there a clear statement of need explaining what problems the software is designed to solve and who the target audience is? NoAdditional CommentsThe problem statement is addressed in the introduction, which mentions the need for a GUI tool as a much more accessible way to edit the XML-based model syntax.   However, it is somewhat confusing who exactly the intended audience of the paper is. Is the paper targeted at researchers that already use PhysiCell, but might want to switch to the GUI version? Or should it (also) target the potential new user-base of researchers interested in using ABMs, for whom the XML version was not sufficiently accessible and who will now gain access to these models because there is a GUI?  Specifying the intended audience might impact some sections of the paper. For example, for users who already use PhysiCell, the step-by-step tutorials might not be useful since they would already know most of the available options; they would just need a quick overview of what info is in which tab. But if the paper is (also) targeted at potential new users, then some additional information could make both the paper and the tool much more accessible, such as:  - A clear comparison to other modeling frameworks and their functionalities. Why should they use PhysiCell instead of one of the other available (GUI) tools? For example, the referenced Morpheus, CC3D and Artistoo all focus on a different model framework (CPMs); this might be worth mentioning. And what about Chaste? Does it represent different types of models, or are there other reasons to consider PhysiCell over Chaste or vice versa? For new users, this would be important information to include. The paper currently also does not mention other frameworks except those that offer a GUI. While the main point of the paper is the addition of the GUI, for completeness sake it might still be good to mention a broader overview of ABM frameworks and how they compare to PhysiCell, or simply to refer to an existing paper that provides such an overview. - The current tutorial immediately dives into very specific instructions (what to click and exact values to enter), often without explaining what these options mean or do. New users would probably appreciate to get a rough outline of which types of processes can be modelled, and which steps they would take to do so. This could be as easy as summarising the different main tabs before going into the details. I understand that some of these explanations will overlap with the main PhysiCell software – but considering that the GUI will open up modelling to a different type of community, it might make sense to outline them here to get a self-contained overview of functionality. - Indeed, if the above information is provided, the detailed tutorial might fit better as an appendix or in online documentation. That would also leave more space to explain not only which values to enter, but also what these variables do, why choose these values, what other options to consider, etc. Having this information together in one place would be very useful for beginning users. 
Is the source code available, and has an appropriate Open Source Initiative license <a href="https://opensource.org/licenses" target="_blank">(https://opensource.org/licenses)</a> been assigned to the code?YesAdditional CommentsThe software is available under the GPL v3 licence.As Open Source Software are there guidelines on how to contribute, report issues or seek support on the code?YesAdditional CommentsThere is a Github repository, ensuring that it is possible to contribute and report issues, and the paper explicitly invites community contributions. However, although the paper mentions that it is possible to seek support through Github Issues and “Slack channels”, we could find no link to the latter resource. This should probably be added to make this resource usable for the reader (or otherwise the statement should be removed).Is the code executable?YesAdditional CommentsIs installation/deployment sufficiently outlined in the paper and documentation, and does it proceed as outlined?NoAdditional CommentsMostly yes, as installation and deployment are outlined in the paper and documentation. However, we did notice a couple of issues:  - The studio guide explains how to compile a project in PhysiCell (https://github.com/PhysiCell-Tools/Studio-Guide/blob/main/README.md), but does not mention that Mac users need to specify the g++ version at the top of the Makefile. This is explained in a separate blog (http://www.mathcancer.org/blog/setting-up-gcc-openmp-on-osx-homebrew-edition/) but should be outlined (or at least referenced) here as well. - There are several different resources covering the installation process, referring to e.g. github.com/physicell-training, github.com/PhysiCell-Tools/Studio-Guide, and the abovementioned blog. But this might not be very accessible to all users targeted by the new GUI functionality (especially when command line interventions and manual Makefile edits are involved). While not all of this has to be changed before publication, having all information in one place would already improve accessibility to a larger user-base.  - When following the instructions (https://github.com/PhysiCell-Tools/Studio-Guide/blob/main/README.md), “python studio/bin/studio.py -p -e virus-sample” the -p flag gives an error: “Invalid argument(s): [‘-p’]”. We assumed it has to be left out, but perhaps the docs have to be updated.
Is the documentation provided clear and user friendly?YesAdditional CommentsMostly yes, as there is already a lot of documentation available. However, the user-friendliness could be improved with some minor changes.  For example, the documentation could be made more user-friendly if resources were available from a central spot. Currently, information can be found in different places: - https://github.com/PhysiCell-Tools/Studio-Guide/blob/main/README.md provides installation instructions and a nice overview of what is where in the GUI, but as mentioned above, does not mention potential issues when installing on MacOS. - The paper provides very detailed examples; these might be nice to include along with the abovementioned overview. - Potentially other places as well. It would be great if the main documentation page could at least link to these other resources with a brief description of what the user will find there.   Further, some additions would make the documentation more complete: - It would be good to have an overview somewhere of all the configuration files that can be supplied/loaded (e.g. those for “rules” and for initial configurations). - A clearer instruction/small tutorial on how to use simularium and paraview with physicell studio; especially for paraview there is no instruction on how to use your own data or make your own `.pvsm` file  In the longer term, it might be worthwhile to set up a self-contained documentation website (this is relatively easy nowadays using e.g. Github pages), which can outline dependencies, installation instructions, a quick overview, detailed tutorials, example models, links to Github issues/slack communities. This is not a requirement for publication but might be worth looking into in the future as it would be more user-friendly. 
Is there enough clear information in the documentation to install, run and test this tool, including information on where to seek help if required?YesAdditional CommentsWhile the paper mentions the option of further support via Github issues and slack, it would be good to list these options in the Github documentation as well. In addition, it might be a good idea to include a tab “help” in the GUI tool that provides links to the Github Issues and/or slack communities. Not all users of the tool (especially given the new option with the GUI) will be familiar enough with Github to search for the Issues page themselves, so adding the link would be a minor change that could improve user experience. Is there a clearly-stated list of dependencies, and is the core functionality of the software documented to a satisfactory level?NoAdditional CommentsThe core functionality of the software is nicely outlined in the Github README (https://github.com/PhysiCell-Tools/Studio-Guide/blob/main/README.md), but as mentioned before, this high-level overview is missing in the paper itself.   The README and paper recommend installing the Anaconda python distribution to get the required python dependencies. This is fine, but adding a setup file or requirements.txt might still be useful for users who are more familiar with python and want a more minimal installation.   Providing a conda environment.yml that allows running the studio along with paraview and/or simularium might also be helpful. Note that running the studio with simularium in anaconda did not work because anaconda did not have the required vtk v9.3.0; instead we had to install simularium without anaconda (“pip3 install simularium”).
Have any claims of performance been sufficiently tested and compared to other commonly-used packages? Not applicableAdditional CommentsIs test data available, either included with the submission or openly available via cited third party sources (e.g. accession numbers, data DOIs)?YesAdditional CommentsAre there (ideally real world) examples demonstrating use of the software? YesAdditional CommentsThe detail tutorial nicely walks the reader through the tool (although as mentioned before, a high-level overview is missing and the level of detail feels slightly out of place in the paper itself).  When walking through the example in the paper and the supplementary, we did run into a few (minor) issues: - It might be good to stress explicitly that after copying the template.xml into tumor_demo.xml, the first step is always to compile using “make”. The paper mentions “Assuming … you have compiled the template project executable (called “project”) …”. But it might not be immediately clear to all users how exactly they should do so (presumably by running “make tumor_demo” after copying the xml file?). - When running “python studio/bin/studio.py -c tumor_demo.xml -e project” as instructed, a warning pops up that “rules0.csv” is not valid (although the tool itself still works). - The instructions for plotting say to press “enter” when changing cmin and cmax, but Mac offers only a return key. Pressing fn+return to get the enter functionality also does not work; it might be good to offer an alternative for Mac. - When reproducing the supplementary tutorial, results were slightly different. It might be good if the example would offer a random seed so that users can verify that they can reproduce these results exactly. In our hands, when reproducing figs 39, 40, 48, 49 yields way more (red) macrophages (even when running multiple times), but we could not be sure if this is due to variation between runs, or a mistake in the settings somewhere. 
Additional CommentsThe paper mentions that they have started setting up automated testing, but it does not give an idea of what the current test coverage is. Did they add a few tests here and there, or start to systematically test all parts of the software? I understand the latter might not be achievable immediately, but it would be good if users and/or contributors can at least get a sense of how good the current coverage is.   (Note: the framework uses pytest, which seems to offer some functionality to generate coverage reports, see e.g. https://www.lambdatest.com/blog/pytest-code-coverage-report/).  The code in studio_for_pytest.py has a comment “do later, otherwise problems sometimes”, but it is not entirely clear if the relevant issue has been resolved.
Any Additional Overall Comments to the AuthorThe presented tool offers a GUI interface to the PhysiCell framework for agent-based modeling. As outlined for the paper, this offers significant value to the users since editing a model is now much more accessible. The tool comes with extensive functionality and instructions. Overall, the tool functions as advertised, and will be of great value to the community of PhysiCell users that now have to edit XML files by hand. It is therefore (mostly) publishable as is if some of the issues with installation (mentioned above) can be straightened out.  That said, we do think some improvements could make both the tool and the paper more accessible to a larger user audience. Most of these have been mentioned in the other questions, but we will list some additional ones below. Note that many of these are just suggestions, so we will leave it up to the authors if and when they implement them.   Suggestions for the paper:  While the paper nicely outlines design ideas and usage of the tool, there were some points where we felt that the main point did not quite come across, for example: - As mentioned in the question about problem statement and intended audience, adding some information to the paper would make it a more useful resource to users not yet familiar with PhysiCell (see remarks there). - The section “Design and development” describes the development history of the tool. In principle this is a valuable addition, because it illustrates how the project is under ongoing development and has already been improved several times based on feedback of users. However, the amount of information on each previous stage is slightly confusing; it is not entirely clear how this relates to the paper and current tool. If the main point is to showcase that the current tool has been built based on practical user experiences, this would probably come across better if this section was somewhat shorter and focused on the design choices rather than previous versions. If the main point is something else, it should be clarified what the main idea is. - The point of Table 1 was unclear to us – consider removing or explaining the main idea. - Several figures do not have captions (e.g. Figure 1 but also others); it would be helpful to clarify what message the figure should convey.  - P4 “adjust the syntax for Windows if necessary” – is it self-explanatory how users should adjust? Consider adding the correct code for windows as well if possible, since users that want to use the GUI tool might not be familiar with command line syntax. - P6 “if you create your own custom C++ code referring directly to cell type ID” – this functionality is never discussed. This might be part of the general PhysiCell functionality, but it would be good to at least provide a link to a resource on how you could do this. - P8 “Only those parameters that display … editing the C++ code” – it was not entirely clear to me what this means, could you clarify? - P13 mentions you can immediately see changes to the model parameters made. This is very useful for prototyping when users want immediate feedback. However, what happens when you try to save output for a simulation where parameters were changed while the simulation was running? Would users be reminded that their current output is not representative? - Discussion: it is good to mention that the tool is already being used. Can you give an indication based on your experience how long it takes new users to learn to navigate the tool? This might be useful information to add in the paper. - The last statement on LLMs seems to come out of nowhere. Consider leaving it out or expanding further on what would be needed to make this work/how feasible this is.  Further comments on the tool itelf:  - The paper mentions that results may not be fully reproducible if multiple threads are used (I assume this is the case even when a random seed is set). In this case, would it make sense to throw a warning the first time a user tries to set a seed with multiple threads, to avoid confusion as to why the results are not reproducible? - Unusable fields are not always greyed out to indicate that they are disabled, which sometimes makes it seem as though the tool is unresponsive. In other places unusable options are set to grey, so it might be good to double-check if this is consistent. - At the initial conditions (IC) page there is no legend; it might be good to add one. - There are some small inconsistencies between the field names mentioned in the paper and those in the tool/screenshots. For example “boundary condition” (p5) should be “dirichlet BC”, “uptake” (p6) should be “uptake rate”. For the latter, the paper mentions that the length scale is 100 micron but this should be visible in the tool as well. - Not all fields have labels, so it is not always clear what the options do (see e.g. drop-downs in Figure 6).  - There are a few points in the tool where you have to “enable” a functionality before it works, but this might not always be intuitive. For example, if you upload a file with initial conditions, it can be assumed that you want to use it. There might be good reasons for this in some cases but in general, consider if all these checkpoints are necessary or if this could be simplified. Same goes for the csv files that have to be saved separately instead of through the main “save” button – in the long term it might be worth saving all relevant files when they are updated, or at least throwing a warning that you have to save some of them separately.  
RecommendationMinor Revisions